# The Potential Antiviral Effects of Selenium Nanoparticles and Coated Surfaces

**DOI:** 10.3390/antibiotics11121683

**Published:** 2022-11-23

**Authors:** Jonathan Kopel, Joe Fralick, Ted W. Reid

**Affiliations:** 1School of Medicine, Texas Tech University Health Sciences Center, Lubbock, TX 79430, USA; 2Department of Immunology and Molecular Microbiology, Texas Tech University Health Sciences Center, Lubbock, TX 79430, USA; 3Department of Ophthalmology and Visual Sciences, Texas Tech University Health Sciences Center, Lubbock, TX 79430, USA

**Keywords:** selenium, selenium nanoparticles, metals, viruses, coating

## Abstract

Modern epidemics quickly spread across borders and continents with devastating effects on both human health and the world economy. This issue is made worse by the various ways that infections are spread, including through aerosol, droplets, and fomites. The antibacterial qualities of various surface materials and coatings have been the subject of much research. However, the antiviral activity of metal coatings can be heavily influenced by imbalances in metal distribution and the presence of other metal impurities. As such, there is interest in developing novel surface coatings that can reduce the transmission of active viral particles in healthcare facilities. In recent years, the non-metals, such as selenium and nanoparticles, have acquired greater interest from the medical and scientific community for their antiviral surface activity. In this review, we will discuss the cellular and physiological functions of selenium in mammalian cells and against viral infections. We then discuss the mechanism behind selenium coated surfaces and their efficacy against bacterial infections. Lastly, we examine the antiviral activity of selenium, and the potential antiviral activity of selenium nanoparticles and coatings.

## 1. Introduction

Modern pandemics have been a persistent problem throughout human history leading to disruptions to human health and the world economy. In recent years, this issue has worsened with the emergence of contagious infections that spread through several methods including, through aerosol, droplets, and fomites. Historically, the usual method for reducing the spread of bacterial and viral pathogens has included multiple-barrier protection (e.g., personal protective equipment), strict hygiene standards, and immunization programs [1]. The antibacterial qualities of various surface materials and coatings have been the subject of much research. Many surface materials have both antiviral and antimicrobial capabilities. Some of the novel surface coating methods to reduce viral particle transmission have included passive pathogen-repellent surfaces, surface-bound active antimicrobials, and biocidal coatings. One common surface coating method includes metal coated surfaces, such as silver, gold, and cobalt [2,3]. However, the antiviral activity of metal coatings can be heavily influenced by imbalances in metal distribution and the presence of other metal impurities. Furthermore, there are significant variations in how different bacteria and viruses react to different prophylactic and therapeutic methods [1]. As such, there is interest in developing novel surface coatings and novel antiviral substances, such as nanoparticles, that can reduce the transmission of active/infective viral particles [4,5,6,7,8,9,10,11,12,13,14,15,16,17,18,19].

In recent years, the non-metals, such as selenium, have acquired greater interest from the medical and scientific community for their antiviral activity using nanoparticles or selenium coated surfaces [19,20]. Due to their unique physical and chemical properties, nanoparticles have been the subject of extensive research over the past few decades. Nanomaterials’ primary characteristics are mostly determined by their shape and particle size, which is between 1 and 100 nanometers (nm) in diameter [19,20]. Given that selenium (Se) is an essential element with numerous functions in the human body, including protection of the cardiovascular and liver organs, selenium nanoparticles (SeNPs) have attracted significant interest for their unique properties to treat several different disease processes [19,21,22,23]. In this review, we will discuss the cellular and physiological functions of selenium in mammalian cells and against viral infections. We then discuss the mechanism behind selenium coated surfaces and their efficacy against bacterial infections. Lastly, we examine the antiviral activity of selenium, and the potential antiviral activity of selenium nanoparticles and coatings.

## 2. Physiological Functions of Selenium

Inorganic minerals, such as selenide, selenate, and selenite, are abundantly dispersed in the crust of the Earth [24]. Virtually all living things require selenium for optimum health [25]. Selenium is found in biological systems in the forms of selenocysteine, dimethyl selenide, selenomethionine, and selenomethylcysteine [26,27,28,29]. These diverse forms of selenium are integrated as selenocysteine in enzymes used to control several biological functions, such as the immune system response and thyroid function [29,30,31]. In particular, selenoproteins are important components of the cellular antioxidant machinery, which scavenge free oxygen radicals to protect tissue against oxidative damage [32,33]. Among the 25 discovered human selenoproteins, glutathione peroxidase has gained widespread acclaim for catalyzing the oxidation of monomeric glutathione into glutathione disulfide and the reduction of hydrogen peroxide into water by using selenocysteine as a prosthetic group [34,35].

In addition to its antioxidant activity, recent studies examining selenium have shown its importance in preventing the spread of viral infections within the body. The majority of selenium’s health benefits result from the incorporation of selenocysteine into selenoproteins [36]. The UGA codon, which is often used as a signal to stop protein synthesis, is used to encode selenocysteine. Specifically, a selenium atom substitutes a sulfur atom in the structure and function of selenocysteine, giving it increased catalytic activity [36]. The following sub-families of selenoproteins have been thoroughly characterized: glutathione peroxidase, thioredoxin reductases, methionine sulfoxide reductase, and selenoproteins in the endoplasmic reticulum. Selenoproteins are essential for many molecular pathways, including those involving oxidative stress response, redox regulation, mitochondrial function, endoplasmic reticulum stress, and immune and inflammatory responses against viral infections [37]. Cross-talk between these pathways are crucial to produce a sufficient response to viral infections [37]. In addition, sufficient selenium in the body also maintains the structural and functional integrity of pulmonary epithelial barriers since selenoproteins reduce lipid and phospholipid hydroperoxides to protect epithelial barriers from reactive oxygen species (ROS) [37]. Similarly, antioxidant selenoproteins shield neutrophils from inward oxidative stress. As such, it is not surprising that low selenium levels has been associated with a worse response to viral infections.

## 3. Selenium Deficiency and Viral Infections

Viruses with an affinity for the respiratory tract, such as the influenza virus, adenovirus, and respiratory syncytial virus cause a wide range of symptoms in patients, including nasal congestion, sore throat, fever, and severe acute respiratory distress syndrome [38]. When an individual is infected with a virus, the viral particle effects the body’s oxidant state, disrupting the equilibrium between prooxidants and antioxidants. Specifically, viruses cause the production of reactive oxygen species (ROS)-producing enzymes, such as xanthine oxidase (XO) and nicotinamide adenine dinucleotide phosphate oxidases (NADPH oxidases, Nox), to decrease. This reduces the reduction capacity of the cell, leading to irreversible cell and tissue death [38]. This rise in ROS and consequent oxidative stress from viral infections create an environment that is conducive for viral replication [39]. Superoxide dismutase, catalase, peroxiredoxins, and glutathione peroxidases are antioxidant enzymes that play a significant part in reducing the oxidative stress secondary to viral infections [40]. Activation of inflammatory pathways results in an increase in cytokine production and consequent tissue damage after an increase in ROS from viral infections. The nuclear factor κB (NF-κB) signaling is activated by the elevated ROS during viral infections, which increases the buildup of inflammatory cytokines. These inflammatory cytokines are produced in excess, leading to a cytokine storm that severely damages major organs in the body, such as the lungs [41].

Tas, using murine in vivo studies, found that selenium deficiency decreased glutathione peroxidase activity and facilitated the emergence of viral variants with greater pathogenicity in patients carrying the human immunodeficiency virus (HIV) [42]. It is hypothesized that increased oxidative stress, brought on by the lack of glutathione peroxidase, causes a higher severity of viral infections in selenium-deficient hosts [29]. When compared to infected persons with adequate selenium levels, hospitalized HIV-infected patients with low selenium levels have a higher mortality rate. Additionally, selenium supplementation increased glutathione peroxidase activity and, subsequently, reduced HIV-1 activation [43]. In addition, a deficiency in selenium is associated with worsening progression and pathogenesis of several other viruses, including coxsackie virus, influenza, hepatitis B and C, poliovirus, and Severe Acute Respiratory Syndrome Coronavirus 2 (SARS-CoV-2) [44]. Overall, the importance of selenium against a wide range of viral infections makes selenium an attractive for antiviral coatings given its versatility and low cost. In the subsequent sections, we propose a mechanism by which selenium coatings eliminate viral particles and current studies showing their effectiveness. 

## 4. Proposed Mechanism of Antiviral Activity using Selenium Coatings

These organo-selenium compounds kill bacteria by means of selenium’s ability to catalyze the formation of superoxide radicals. For example, previous studies on selenoesters and selenoanhydrides have shown these selenium compounds are effective antiviral agents against the herpes simplex virus [45]. Superocide radicals are formed from oxygen by organo-selenium compounds from oxygen using reduce glutathione, present in saliva, sweat, tears, blood and other body secretions according to the following reactions: R-Se^−^ + 2G-SH +O_2_^−^ > R-Se + G-SS-G + O_2_^−^∙) [46]. In acidic microenvironments, such as the Gouy-Chapman interface of a cell or virus’s phospholipid membrane, the negatively charged superoxide radical (O_2_^−^∙) is readily protonated to become the uncharged and more reactive perhydroxyl radical, HOO∙. It is hypothesized that perhydroxyl radicals can pass into membranes and covalently modify its unsaturated lipids, causing local disruption and permeabilization of the lipid bilayer membrane [46]. The cell killing effect by selenium-conjugated polymers requires close contact (less than a micron) between the polymer surface and the bacteria. This is because the oxygen radicals decay rapidly into non-toxic compounds [46]. For this reason, the polymer is effectively non-toxic to cells and tissues that are not in intimate contact with the polymer. An example of this application of selenium coated surfaces is shown in Figure 1. In addition, these organo-selenium compounds can be designed to be target specific against the microorganism of interest [47].

There is ongoing interest in using selenium’s antimicrobial properties to manufacture medical devices like wound bandages, catheters, contact lenses, dental sealants, and tympanostomy tubes to reduce the development of bacterial biofilms and prevent tissue infections [48,49,50]. In these devices, the organo-selenium (OS) monomer coating is applied to the devices’ interior or exterior to eliminate bacteria from entering the patient. One study found that the surface colonization of laboratory and clinical strains of *S. aureus*, *P. aeruginosa*, and *S. epidermidis* was dramatically reduced when OS coating was applied to the surface of the gauze bandage [51]. The results were confirmed in vivo since no bacterial biofilm could grow on the OS-coated bandage or in the lesion it covered. Furthermore, the OS coating in the bandage maintained complete or partial inhibitory efficacy after being stored for 6 years at room temperature, for 1 month at 37°C in aqueous solution, and for 15 min in boiling water [51]. Similar outcomes were obtained when the OS monomers were polymerized into a coating on a polyester bandage. After a month at 37°C in PBS, this bandage remained stable [48]. It was proposed that the blending of OS monomers into the polyester polymer might increase the stability of the OS coated bandage, reducing the usual issues of rip, surface wear, and leaching [48].

Similarly, selenocyanatodiacetic acid (SCAA)-coated hemodialysis catheters did not form *S. aureus* biofilms [52]. Furthermore, for a sustained length of time through the catheter, biofilm development was suppressed under flow parameters that mimicked the host blood flow [52]. Strong covalent bonds were used to fix selenium to the catheter’s surface [52]. The fact that this quantity of released selenium had no impact on the survival of the animals or on cell shape, attachment, or density suggests that the SCAA coating on the catheters was safe to use [52]. Dental pit and fissure sealants were also used to investigate the effect of OS coating on the prevention of oral bacteria biofilm development [53]. SeLECT-DefenseTM, a sealant with a 1% selenium coating, totally prevented the growth of *S. salivarius* biofilms. After 2 months of soaking in PBS at 37 °C, the sealant still had an inhibiting effect [53]. These findings were translated into a clinical study with 120 adolescents who were predisposed to developing caries [54]. The OS coated pit and fissure sealant (DenteShield^TM^) demonstrated zero plaque development and a 100% caries prevention in comparison to the control sealant, which shown 7% and 12% plaque formation nine and twelve months after the trial’s commencement. No negative impacts were seen during the use of the DS sealant [54].

Using colony forming unit (CFU) measurement, Confocal laser scanning microscopy, and COMSTAT imaging, Wang et al. coated Donaldson tympanostomy tubes with Seldox, a methacrylate containing covalently attached selenium, and examined the bacterial growth, colonization, and biofilm formation on the surface [50]. They reported that OS coat reduced the growth of *M. catarrhalis,* nontypable *H influenzae*, and *S. aureus* biofilms [50]. The OS coating of wound dressings, catheters, contact lenses, dental sealants, and tympanostomy tubes was therefore shown to be effective in preventing the formation of bacterial biofilms while being stable and safe for use. As such, selenium coatings have become an attractive method for developing antibacterial and, recently, antiviral surface coatings. Overall, the importance of selenium against a wide range of viral infections makes selenium an attractive for antiviral coatings given its versatility and low cost. In the subsequent sections, we propose a mechanism by which selenium coatings eliminate viral particles and current studies showing their effectiveness. In the following section, we explore some of the recent applications of selenium nanoparticles and coated surfaces against viral particles.

## 5. Antiviral Activity of Selenium Nanoparticles

Selenium nanoparticles are effective antiviral substances against viral particles (Table 1). After delivering SeNPs to HepG2 liver cell lines infected with hepatitis B (HBV), proinflammatory markers tumor necrosis factor (TNF) and tumor growth factor (TGF) levels decreased 1.5 and 1.3 times, respectively; in addition, SeNPs also reduced the levels of IL-8 and IL-2 by 46% and 43%, respectively [20]. All proinflammatory markers were much lower after receiving SeNPs treatment than in the control group. Selenium nanoparticles also decreased DNA fragmentation in HepG2 infected cell lines. Specifically, the study showed an 8-fold increase in the tail length and a 3.7-fold increase in tail DNA percentage in the HepG2 cell lines infected with HBV [20]. When HeG2 cells infected with HBV were treated with SeNPs, DNA damage also decreased [20].

In addition, SeNPs were also found to be effective against enterovirus 71 (EV71), which is the main cause of hand, foot, and mouth disease [55]. A study by Lin et al. showed that SeNPs attached with small interfering RNAs (siRNA) targeting the EV71 VP1 gene, the surface of the siRNA was coated with polyethylenimine (PEI) to form a selenium nanoparticle—polyethylenimine—siRNA complex (Se@ PEI@siRNA) [55]. In the nerve cell line, SK-N-SH, the Se@PEI@siRNA showed a superior efficacy preventing the nerve cells from being infected. Specifically, Lin et al. showed that the Se@PEI@siRNA prevents the replication of EV71 by reducing the amount to which SK-N-SH cells in the sub-G1 phase [55]. A recent study examined the effectiveness of chitosan-coated selenium nanoparticles (CS-SeNPs) at inhibiting the replication of the porcine reproductive and respiratory syndrome virus (PRRSV) [56]. The study found that the treatment with CS-SeNPs greatly reduced the oxidative stress brought on by r-PRRSV-green fluorescent protein (GFP) infection. Specifically, this caused by an increase in glutathione peroxidase activity and glutathione formation as well as preventing H_2_O_2_ generation. At 24 and 48 h after infection, the CS-SeNPs treatment also significantly decreased viral titers of r-PRRSV-GFP in Marc-145 cells [56]. In addition, the CS-SeNPs inoculation substantially reduced the rise in apoptosis rates brought on by r-PRRSV-GFP infection through decreasing ROS production, c-Jun N-terminal Kinase phosphorylation levels, and cleavage of caspase-3 and Poly (ADP-ribose) polymerases [56]. The study demonstrated antiviral activity of Se on the surface of Cs-SeNPs but not necessarily on the surface of other objects. A similar study found that SeNPs inhibit H1N1 infection of Madin-Darby canine kidney (MDCK) cells through impeding chromatin condensation and DNA fragmentation [57]. In MDCK cells, SeNPs dramatically reduced the generation of reactive oxygen species and, subsequently, inhibit apoptotic processes induced by H1N1 virus infection by increasing the level of glutathione peroxidase 1. SeNPs improved the amount of glutathione peroxidase 1 in MDCK cells, which in turn prevented the apoptosis brought on by H1N1 virus infection [57]. The effectiveness of SeNPs against H1N1 was further enhanced when the anti-influenza drugs, oseltamivir, amantadine, ribavirin, was attached to the surface of the SeNP [58,59,60].

Other studies have shown the SeNPs are effective against avian flu viruses. A study by Yehia et al. showed that SeNPs increase cellular immunity and decrease inflammation to improve vaccine effectiveness and offer superior defense against Highly Pathogenic Avian Influenza A H5N1 (HPAI-H5N1) in chickens [61]. Specifically, homologous whole inactivated H5N1 vaccine protection was improved in a dose-dependent manner with SeNPs incorporated in the hens’ food and vaccine formulation. The authors aruged that the improved protectivity brought on by SeNPs was associated with its anti-inflammatory effects and improvements to cellular immunity [61]. SimilarlyA study by Najjari et al. also found that hexanic extracts of fig (*Ficus carica*) and olive (*Olea europaea*) fruit attached to SeNPs inactivated avian influenza virus subtype H9N2 and improved the vaccine effectiveness [62].

Another study examined the use of Thujaplicin attached to selenium nanoparticles. In a study by Wang et al., functionalized SeNPs with enhanced antiviral activity were created by surface-modifying Se attached Thujaplicin (Se@TP) [63]. The study found that the cell survival rate of MDCK cells was 45% higher when treated with Se@TP compared to the viral control group [63]. Furthermore, the Se@TP prevented DNA fragmentation, chromatin condensation, and H1N1 infection in MDCK cells. Evidently, MDCK cells were unable to produce reactive oxygen species in the presence of Se@TP [63]. Additionally, Se@TP protects mice with H1N1 infection from lung damage through in vivo eosin and hematoxylin staining studies [63]. The study’s findings demonstrate that Se@TP has lower toxicity and superior antiviral properties to eliminate the H1N1 influenza virus. Subsequent analysis found that Se@TP prevented caspase-3-mediated apoptosis by generating ROS [63]. Furthermore, Se@TP reduced apoptosis in MDCK cells through control of the AKT and p53 signaling pathways. Using these results, Wang et al. argued that the nanosystem of Se@TP may offer a potential selenium species with antiviral action against the H1N1 influenza virus [63]. In these examples, the SeNPs enter infected cells to prevent viral replication. However, a recent study found that SeNPs may inactivate viral particles if attached on surfaces in a mechanism that is similar to how selenium coatings inactive bacteria.

## 6. Antiviral Activity of Selenium Coated Surfaces

In another study, printing technology was used to integrate selenium nanoparticles with polyester textiles to create materials with multifunctional capabilities, such as combination antiviral and antibacterial activity as well as coloring [64]. Tensile strength and color fastness of the printed polyester textiles containing selenium nanoparticles were among the measured variables. Utilizing transmission electron microscopy and scanning electron microscopy, the selenium nanoparticles revealed excellent uniformity and stability with diameters ranging from 40–60 nm and 40–80 nm [64]. Using the selenium nanoparticles, a significant level of disinfection activity of 87.5% was also observed against the SARS-CoV-2 coronavirus. In addition, the selenium coated nanoparticles on the fabric exhibited little toxicity when exposed to mammalian cell lines. These studies suggest that selenium coatings and metal surfaces may act as effective antiviral agents against current and future viruses. Compared to the previous examples of selenium nanoparticles, the selenium nanoparticle coatings generated superoxide radicals that inactivated the viruses, which is an important property of Se coated surfaces.

## 7. Conclusions

Selenium coatings remain an attractive method for reducing the spread of bacteria and viruses. There is a wide range of substances with antiviral and antibacterial activities. There are several alternatives when novel chemistries are designed and engineered, which is another possibility. These materials’ antimicrobial characteristics have received a lot of attention, but there are much less data on their antiviral activities, which is a gap that needs to be filled. It is obvious that material science may contribute significantly to the creation of theoretical and actual strategies to contain infectious outbreaks. It is important to consider both tried-and-true and novel broad-spectrum antiviral tactics because they might help us better prepare for and face the challenge of upcoming viral pandemics.

## Figures and Tables

**Figure 1 antibiotics-11-01683-f001:**
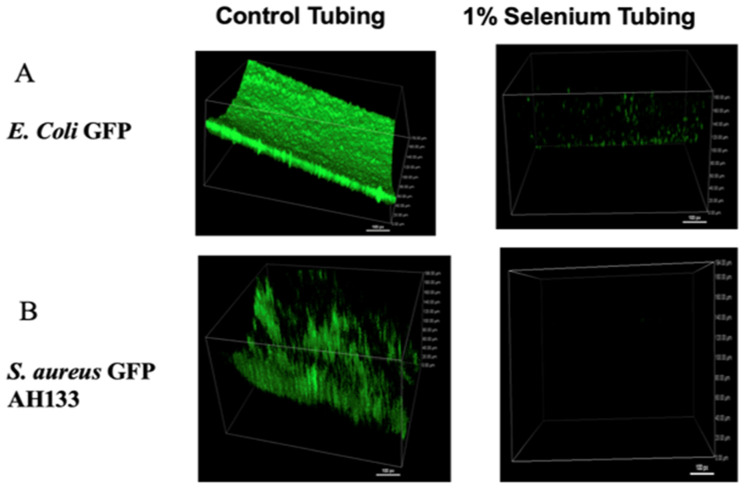
Confocal laser scanning microscopy of biofilm formed by *E. coli* GFP (**A**) and *S. aureus* GFP AH133 (**B**) on control tubing (left) and selenium compound polymerized tubing (right). Images were acquired at 2 µm intervals through the biofilms using a Nikon A1+/AIR+ Confocal Microscope.

**Table 1 antibiotics-11-01683-t001:** Selenium Nano-particles and Coated Surfaces against Viral Particles.

Selenium Nanoparticles	Compound Attached	Targeted Virus
Gad et al. [20]	None	Hepatitis B
Lin et al. [55]	Polyethylenimine—siRNA	Enterovirus 71
Shao et al. [56]	Chitosan	Porcine reproductive and respiratory syndrome virus
Liu et al. [57]	None	H1N1 Influenza Virus
Li et al. [58]	Oseltamivir	H1N1 Influenza Virus
Lin et al. [59]	Ribavirin	H1N1 Influenza Virus
Li et al. [60]	Amantadine	H1N1 Influenza Virus
Yehia et al. [61]	None	H5N1
Najjari et al. [62]	Hexanic extracts	H9N2 Influenz Virus
Wang et al. [63]	Thujaplicin	H1N1 Influenza Virus
**Selenium Coated Surface**	**Compound Attached**	**Targeted Virus**
Abou et al. [64]	Selenium Nanoparticles	SARS-CoV-2

## Data Availability

Not applicable.

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
