# Peer review of "The Potential Antiviral Effects of Selenium Nanoparticles and Coated Surfaces"

_antibiotics, 2022, doi:10.3390/antibiotics11121683_

Round 1

Reviewer 1 Report

The authors reviewed the antiviral effects of selenium coatings. The submission could be accepted after revision taking into account the following points:-

1.      The review is very short and a comprehensive discussion should be added.

2.      The figures are naïve and should be improved. The authors can reprint Figures from the literature with permission.

3.      The literature should be summarized in a Table summarizing the literature.

4.      References should be updated to be broad of interest topic. I suggest these References; https://doi.org/10.1007/s41204-021-00109-0; https://doi.org/10.1016/j.cej.2021.132966;  https://doi.org/10.3390/nano11071788; https://doi.org/10.3390/chemosensors10070287

5.      The language should be revised, and typos should be corrected.

Minors

6.      Correct typos such as ‘O2-*’, 

Author Response

Reviewer #1
1.      The review is very short and a comprehensive discussion should be added.

We added more content and discussion to this manuscript. We expanded on the physiological functions of selenium and the function of selenium against viruses. In addition, we added citations on the antimicrobial effect of selenium coated surfaces. We also added additional citations on selenium nanoparticles and selenium coated nanoparticles.

  1. The figures are naïve and should be improved. The authors can reprint Figures from the literature with permission.

We appreciate the reviewer’s comment. We decided to change the figures as suggested to included a novel example of a selenium coated surface against bacteria in pipes. We wanted to show the effectiveness against microorganisms before continuing to discuss it with regard to viruses.

  1. The literature should be summarized in a Table summarizing the literature.

We added a table that summarized the selenium nanoparticles and the selenium nanoparticle surfaces to the manuscript. We put this as Table 1.

  1. References should be updated to be broad of interest topic. I suggest these References; https://doi.org/10.1007/s41204-021-00109-0; https://doi.org/10.1016/j.cej.2021.132966; https://doi.org/10.3390/nano11071788; https://doi.org/10.3390/chemosensors10070287

We added a separate section to the manuscript using the following citations to expand the literature. Specifically, we added the above papers to the citations and also expanded on selenium nanoparticles in the introduction and later on in the manuscript.

  1. The language should be revised, and typos should be corrected.

We appreciate the reviewer’s comment. We made sure to check and update the language of the manuscript. We also made sure to have the other co-authors look over the paper as well.

  1. Correct typos such as ‘O2-*’, 

We made the appropriate change to the manuscript.

Reviewer 2 Report

The review article, as titled, attempts to emphasize on Selenium surfaces and coatings having anti-viral properties that can reduce presence of active viral pathogens and transmission of infection. The review paper focuses minimally on selenium surface coatings and needs significant make over prior to acceptance.

Major comments:

1.      The Abstract is almost the same as Introduction and both sections quoting COVID-19 pandemic introduce need of surface coatings that can reduce presence of active viral pathogens in variety of settings such as hospitals, public transport, and schools; and further state that past studies have discovered novel methods having significantly more anti-bacterial properties and very less anti-viral properties. The remainder of the manuscript focuses minimally on selenium surfaces or coatings demonstrating anti-viral properties but rather on physiology of selenium and selenoproteins, failing to cite references or discuss studies that can demonstrate any anti-viral effects of selenium coatings/surfaces.

2.      Several sections have simply stated selenium coated surfaces showing great anti-viral activity without citing or describing any references. Some examples include lines 39-40, 63-65, and 166-167

3.      The authors have cited 3 studies claiming effectiveness of selenium coatings having anti-viral properties - Ref 59, 60, and 61 (lines 168-169).

Ref 59 describes an in-vitro study where Porcine reproductive and respiratory syndrome virus (PRRSV) infected Marc-145 cells on treatment with Chitosan-coated selenium nanoparticles reduced the PRRSV replication in vitro; and had no description of selenium nanoparticles demonstrating antiviral activity on any kind of surfaces as stated in lines 178-179.

Ref 60, similarly, is also an in-vitro study and describes how selenium nanoparticles inhibits apoptosis induced by H1N1 virus in MDCK cells, and not on selenium coatings or surfaces.

Ref 61, in the entire review paper, is the only correct reference cited which describes polyester fabrics printed with selenium nanoparticles, producing surface coatings with anti-viral properties.

4.      For a review article titled “The Antiviral Effects of Selenium Coatings”, and with Abstract and Introduction emphasizing on selenium surfaces and coatings having anti-viral properties, the body of the review paper minimally justifies the title of the paper.

Author Response

Reviewer #2

The review article, as titled, attempts to emphasize on Selenium surfaces and coatings having anti-viral properties that can reduce presence of active viral pathogens and transmission of infection. The review paper focuses minimally on selenium surface coatings and needs significant make over prior to acceptance.

Major comments:

  1. The Abstract is almost the same as Introduction and both sections quoting COVID-19 pandemic introduce need of surface coatings that can reduce presence of active viral pathogens in variety of settings such as hospitals, public transport, and schools; and further state that past studies have discovered novel methods having significantly more anti-bacterial properties and very less anti-viral properties. The remainder of the manuscript focuses minimally on selenium surfaces or coatings demonstrating anti-viral properties but rather on physiology of selenium and selenoproteins, failing to cite references or discuss studies that can demonstrate any anti-viral effects of selenium coatings/surfaces.

We appreciate the reviewer’s comment. We rewrote large portions of the manuscript and added more relevant details on selenium’s function for viruses and selenium coatings

  1. Several sections have simply stated selenium coated surfaces showing great anti-viral activity without citing or describing any references. Some examples include lines 39-40, 63-65, and 166-167 We appreciate the reviewer’s comment. We rewrote large portions of the manuscript and made sure to attached the relevant citations

  1. The authors have cited 3 studies claiming effectiveness of selenium coatings having anti-viral properties – Ref 59, 60, and 61 (lines 168-169).

Ref 59 describes an in-vitro study where Porcine reproductive and respiratory syndrome virus (PRRSV) infected Marc-145 cells on treatment with Chitosan-coated selenium nanoparticles reduced the PRRSV replication in vitro; and had no description of selenium nanoparticles demonstrating antiviral activity on any kind of surfaces as stated in lines 178-179.

Ref 60, similarly, is also an in-vitro study and describes how selenium nanoparticles inhibits apoptosis induced by H1N1 virus in MDCK cells, and not on selenium coatings or surfaces.

Ref 61, in the entire review paper, is the only correct reference cited which describes polyester fabrics printed with selenium nanoparticles, producing surface coatings with anti-viral properties.

We appreciate the reviewer’s comment. We rewrote large portions of the manuscript and made sure to attached the relevant citations

For a review article titled “The Antiviral Effects of Selenium Coatings”, and with Abstract and Introduction emphasizing on selenium surfaces and coatings having anti-viral properties, the body of the review paper minimally justifies the title of the paper.

We appreciate the reviewer’s comment. We rewrote large portions of the manuscript and made sure to attached the relevant citations. We added more details on selenium coatings and double checked the literature on selenium coatings and antiviral surfaces.

Reviewer 3 Report

The authors have written a review article titled “The Antiviral Effects of Selenium Coatings”. The manuscript must be thoroughly revised, along with some sentence framing, spelling error and grammar check. The manuscript must be corrected with consideration following points.

1.      Abstract first line meaning is not clear

2.      The aim of the review is missing in the abstract

3.      The introduction section needs significant improvement in terms of the past and present status of the topic.

4.      The article is hard to follow, and please revise the manuscript for better understanding to the readers

5.      Since this is a review article, a table must contain that compiles the selenium's antiviral activity and its coating.

6.      How selenium coating is essential for the antiviral activity must be included in the conclusion section.

Author Response

Reviewer #3

The authors have written a review article titled “The Antiviral Effects of Selenium Coatings”. The manuscript must be thoroughly revised, along with some sentence framing, spelling error and grammar check. The manuscript must be corrected with consideration following points.

  1. Abstract first line meaning is not clear

We appreciate the reviewer’s comment. We rewrote the entire abstract

  1. The aim of the review is missing in the abstract

We appreciate the reviewer’s comment. We added a transition focusing on the aim of the review

  1. The introduction section needs significant improvement in terms of the past and present status of the topic.

We appreciate the reviewer’s comment. We rewrote the introduction to be more clear

  1. The article is hard to follow, and please revise the manuscript for better understanding to the readers

We appreciate the reviewer’s comment. We rewrote large portions of the manuscript and made sure to attached the relevant citations

  1. Since this is a review article, a table must contain that compiles the selenium's antiviral activity and its coating.

We added a new figure to the manuscript. The literature on the selenium coatings and antiviral surfaces is relatively new.

  1. How selenium coating is essential for the antiviral activity must be included in the conclusion section.

We appreciate the reviewer’s comment. The conclusion was rewritten as asked. However, selenium coating is not essential for antiviral activity. Rather, the coatings are designed to target microorganisms.

Round 2

Reviewer 2 Report

To every critic comment from my previous review, the authors have simply stated that they have rewritten large portions. This reviewer does not agree with the revisions made. Kindly find comments below:

1.      Significant text highlighted in red is as is taken from version 1, and authors still highlighting in red either suggest the authors are intentionally trying to misguide reviewers or the authors failed to notice that it is the same text as in version 1. Examples are provided below:

a.      Lines 29-43 are same as in version 1 and still have been highlighting red, claiming as new text

b.      Lines 50-53 are same as in version 1 and still have been highlighting red, claiming as new text

c.      Lines 78-95 are same as in version 1 and still have been highlighting red, claiming as new text

2.      New text added from lines 135-196 are all examples of selenium coatings being effective in preventing antibacterial growth, not antiviral. Here the authors are contradicting their own rationale in the Abstract/Introduction that antibacterial qualities of various coating have been the subject of much research. Lines 135-150 includes a procedure for an experiment performed, which is redundant.

3.      Section “Antiviral activity of Selenium coatings” (line 196) still includes the same references from version 1 and comment 3 from my prior response remains unaddressed. These examples are those of invitro studies where treatment by selenium nanoparticles inhibited virus replication. These references do NOT convey the importance or role of selenium coatings having anti-viral properties.

Author Response

  1. Significant text highlighted in red is as is taken from version 1, and authors still highlighting in red either suggest the authors are intentionally trying to misguide reviewers or the authors failed to notice that it is the same text as in version 1. Examples are provided below:
  2. Lines 29-43 are same as in version 1 and still have been highlighting red, claiming as new text
  3. Lines 50-53 are same as in version 1 and still have been highlighting red, claiming as new text
  4. Lines 78-95 are same as in version 1 and still have been highlighting red, claiming as new text

We appreciate the reviewer’s comment. We put in the lines and changes made in this edit to avoid any confusion. We apologize for any confusion, it was not our intention to mislead. We put the descriptions of changes made to the paper below.

Line 1 – paper was changed from review to perspective. The paper is giving a perspective on the use of selenium against viruses based upon previous studies against bacteria and current studies showing the effectiveness of selenium nanoparticles

Line 2 – The title was changed to include selenium nanoparticles and coated surfaces to expand upon the examples of selenium against viruses as well as surfaces coated with selenium nanoparticles

Line 18 – the word “nanoparticles” was added to the manuscript

Lines 20-23 – the order of the paper was clarified. We specifically mentioned the mechanism of selenium against bacterial infections. Since the original studies were done against bacteria, we wanted to show the basis and mechanism behind their use as coated surfaces.

Line 24 – selenium nanoparticles was added to the keywords section

Line 27-28 – this was rewritten to avoid overlap with the abstract

Line 42-43 – we added nanoparticles to this description to set the stage for the discussion on nanoparticles

Line 43 – we added the words active/infective to describe viral particles. We also added the references [4-19] includes references on nanoparticles

Lines 44-52 – we added the description on selenium coated surfaces and nanoparticles to help set the stage for later on in the paper. We also included several new references [19-23] for this very purposes.

Lines 53-54 – we added a summary on the organization of the paper to include nanoparticles against viruses. We also included the discussion on selenium nanoparticles and coatings as well.

Lines 110-111 – changed the sentence to the following: “Tas, using murine in vivo studies, found that selenium deficiency decreased glutathione peroxidase activity and facilitated the emergence of viral variants with greater pathogenicity in patients carrying the human immunodeficiency virus (HIV)”

Lines 136-138 – sentence was rewritten to the following to improve grammar and clarity: “Superocide radicals are formed from oxlygen by organo-selenium compounds from oxygen using reduce glutathione, present in saliva, sweat, tears, blood and othr body sectetions according to the following reactions: R-Se- + 2G-SH +O2- > R-Se + G-SS-G + O2-*)”

Lines 148-149 – sentence was moved to improve clarity and organization

Lines 164-166 – the sentence was rewritten to the following: “It was proposed that the blending of OS monomers into the polyester polymer might increase the stability of the OS coated bandage, reducing the usual issues of rip, surface wear, and leaching”

Lines 194-215 – in this section we added descriptions of nanoparticles against HepG2 cells infected with the hepatitis B virus. In addition, we provided a description of selenium nanoparticles coated with polyethylenimine and siRNA against enterovirus 71.

Line 214 – we included Table 1 which provides descriptions of the selenium nanoparticles, compounds attached to the selenium nanoparticles and the virus they targeted. At the bottom, we included a description of the selenium coated surface with nanoparticles

Lines 233-261 – We provided studies examining the use of selenium nanoparticles against the influenza viruses H5N1. These selenium nanoparticles increased the effectiveness of vaccines and, when attached with different compounds, increased their efficacy against H5N1. In addition, we included a study using thujaplicin attached to nanoparticles against the H1N1 virus. We included a details on the mechanism of action.

Line 262 – we created a separate section to focus on the selenium coated surfaces with nanoparticles to look at the two uses of selenium nanoparticles

Line 293 – we added a description on the acknowledgements as well.

  1. New text added from lines 135-196 are all examples of selenium coatings being effective in preventing antibacterial growth, not antiviral. Here the authors are contradicting their own rationale in the Abstract/Introduction that antibacterial qualities of various coating have been the subject of much research. Lines 135-150 includes a procedure for an experiment performed, which is redundant.

We included the description on the use of selenium against bacteria to show its original use and effectiveness against microorganisms. We then wanted to transition to our description of selenium against viruses through nanoparticles and nanoparticle coated surfaces. It was not to contradict but to transition appropriately to this section. However, if it is not needed, we are willing to remove it. The lines 123-150 were used for the description on figure 1. We removed the description from the paper. 

  1. Section “Antiviral activity of Selenium coatings” (line 196) still includes the same references from version 1 and comment 3 from my prior response remains unaddressed. These examples are those of invitro studies where treatment by selenium nanoparticles inhibited virus replication. These references do NOT convey the importance or role of selenium coatings having anti-viral properties.

We appreciate the reviewers comment. We agree that adding more references distinguishing the uses of selenium nanoparticles. We added several references to look at the use of selenium nanoparticles against viruses. The citation [65] was used to only focus on the use of selenium nanoparticle coated surfaces against the SARS-CoV-2 virus. We thank the reviewer for this suggestion and critique. We found other references that helped clarify the manuscript.

Reviewer 3 Report

Accept

Author Response

We thank the reviewer for their comments!